# Evaluating E-Health Literacy, Knowledge, Attitude, and Health Online Information in Portuguese University Students: A Cross-Sectional Study

**DOI:** 10.3390/ijerph21030271

**Published:** 2024-02-27

**Authors:** Sofia Almeida, Elisabete Pinto, Marta Correia, Nélio Veiga, Armando Almeida

**Affiliations:** 1Centre for Interdisciplinary Research in Health (CIIS), Faculty of Health Sciences and Nursing, Universidade Católica Portuguesa, 4169-055 Porto, Portugal; aalmeida@ucp.pt; 2Centro de Biotecnologia e Química Fina, Escola Superior de Biotecnologia, Universidade Católica Portuguesa, 4169-005 Porto, Portugal; ecbpinto@ucp.pt (E.P.); mmcorreia@ucp.pt (M.C.); 3Centre for Interdisciplinary Research in Health (CIIS), Faculty of Dental Medicine, Universidade Católica Portuguesa, 3504-505 Viseu, Portugal; nveiga@ucp.pt

**Keywords:** e-health literacy, health literacy, university students, COVID-19, online survey

## Abstract

Background: The main objective of the present study was to assess e-health literacy in a sample of Portuguese university students and its association with the level of knowledge and seeking for COVID-19-related information. Methods: This cross-sectional online study was conducted on Portuguese university students. All students completed a questionnaire consisting of demographic characteristics, e-health Literacy Scale (eHEALS), and a questionnaire about knowledge, attitude, and health online information seeking. Results: A total of 534 students (76.8% women), with a mean age of 24.3 years old (SD = 7.8), participated in this cross-sectional study, 53.0% of students were from non-health sciences. The mean score of eHEALS literacy was 28.8 (SD = 5.6). Most students (71.1%) classified the Internet as a useful, or very useful, tool in helping them make health related decisions. The use of the Internet as a tool to research health information for a period of two or more hours (OR = 1.9; CI 95% = 1.2; 3.4), to search online for health information on professional websites (OR = 2.3; CI 95% = 1.4; 3.6), to search in official media (OR = 2.3; CI 95% = 1.4; 3.9), and to study in the field of health sciences (OR = 1.6; CI 95% = 1.1; 2.6) increased the likelihood of having sufficient e-health literacy. Conclusion: From a public health perspective, there is a need to develop programs that increase health literacy among university students.

## 1. Introduction

The concept of health literacy (HL) emerged in the 1990s and is defined as “the extent to which individuals have the capacity to obtain, process, and understand basic health information and available services to make appropriate health decisions” [1]. The importance of health literacy as a research topic has been increasing in recent years and has gained greater importance within the European health agenda [2] and within national policies as a resource for health promotion.

The use of health information technology has led to the emergence of the concepts of e-health and e-literacy. The skills and ability to seek, find, understand, and appraise health information from electronic resources and apply such knowledge to address or solve a health problem is called e-health literacy [3]. According to the authors, e-health literacy encompasses six core competencies: (1) traditional literacy, (2) health literacy, (3) information literacy, (4) scientific literacy, (5) media literacy, and (6) computer literacy (2006). The concept of e-health is complex and involves having various competencies. Thus e-health literacy is influenced by a wide variety of factors such as age, gender, education level, accessibility, and Internet use [4,5,6,7].

International studies such as the European Health Literacy Survey (HLS-EU), in which several European countries participated, provided a concerted understanding of people’s ability to access, understand, analyze, and use health information in the areas of health care utilization, health promotion, and disease prevention [4]. Previous studies in the Portuguese population indicate that HL levels are not very satisfactory. The study of Amaral et al. (2016) [8] indicated that 61% of the surveyed population has a problematic or inadequate level of general health literacy, which indicates the need to promote strategies to increase HL. In the dimension of health promotion, the one that matters most here, 60.2% of the surveyed population had a problematic or inadequate level of health literacy. Currently, the contexts of health information include electronic information sources such as the Internet and social networks.

Portugal, like the rest of the world, was affected by the COVID-19 pandemic, with repercussions in all age groups [9]. The emergence of the COVID-19 pandemic has contributed to a great demand in digital media for information about the new virus, and about health. Digital health tools have become an important resource in the face of this COVID-19 epidemiological situation to be able to navigate online environments and understand the information provided by these media. This current epidemiological context poses new challenges for health literacy and digital health literacy.

University students are an important part of society because they are expected to have a high level of knowledge regarding health and care issues [10,11]. Their knowledge, attitudes, and practices, not only affect their lives, but also exert an influence in society. College students’ level of knowledge, concerning health issues, can be an important determinant of the effectiveness of health education [10,11,12]. Therefore, it is crucial to assess the level of e-health literacy in students and relate it to disease prevention.

The main objective of the present study was to assess the level of e-health literacy in in a sample of university students and relate it to the level of knowledge, attitude, and their pursuit of information using as example the epidemiological situation of COVID-19.

## 2. Materials and Methods

We conducted an observational cross-sectional study involving university students from the Catholic University of Portugal, applying an online questionnaire.

The Catholic University of Portugal is a reference institution in Portugal, with high-level research and strongly connected to society. It has four Campi (Lisbon, Porto, Braga and Viseu) and offers 35 undergraduate degrees, 66 master’s degrees, and 25 doctoral degrees programs.

Data collection occurred from October 2022 to February 2023. All eligible students were invited to participate in the survey, through the institutional email system.

The questionnaires allowed to assess information on participant characteristics, e-health literacy assessment, knowledge, attitudes about COVID-19, and seeking health information sources. The first part evaluated social and demographic variables: gender, age, education, field of study, income, use of the Internet in general health information research, and number of hours of Internet use per day.

### 2.1. E-Health Literacy Assessment

The e-health literacy was measured by the e-health literacy scale (eHEALS), validated for the Portuguese population [12]. The eHEALS scale was developed by Norman and Skinner in 2006 and aims to measure literacy skills use, in assessing the effects of strategies to deliver online information and applications [3]. The eHEALS scale consists of 8 items, where respondents are asked to rate each item on a five-point Likert scale (strongly disagree, disagree, neither agree or disagree, agree, or strongly agree). Total scores range from 8 to 40 with higher scores indicating higher self-perceived eHL. The e-health Literacy Scale includes two supplemental items that measure perceived usefulness of the Internet to make health decisions and perceived importance of accessing health resources on the Internet. eHEALS scores were dived into the threshold values of inadequate (8–20 points), problematic (21–26 points), and sufficient (27–40 points), as well as dichotomized into limited (inadequate + problematic = 8–26 points) and sufficient (27–40 points) [13]. Psychometric testing of eHEALS indicates that it is a valid and reliable instrument [12]. We carried out an internal consistency analysis in this study for the eHEALS scale. Cronbach’s alpha was 0.90, indicating a very good internal consistency.

### 2.2. Knowledge and Attitude Assessment Related to COVID-19

Epidemiological knowledge related to COVID-19 was assessed by five questions, with response options of true, false, and do not know. The questions about knowledge of COVID-19 were based on a literature search [8,9]. Correct answers were scored as 1 and incorrect answers as 0. For each statement, the percentage of correct and incorrect answers was calculated. A knowledge score was calculated with the sum of all questions (ranging from 0 to 5). Attitudes of students to prevent COVID-19 and respiratory infections were evaluated by four questions: having the vaccine was important for my health (they were asked if they considered vaccinations to be important for their health), hygiene care (disinfect hands, avoid rush hours), preventive measures for respiratory infections (wear masks, wash hands). These questions were based on a literature search.

We used a Likert scale of 5 points to access students’ confidence in each preventive/attitude measure (strongly disagree, disagree, neither agree or disagree, agree, or strongly agree). Adequate attitudes corresponded to agree, and strongly agree answers. The score of attitudes ranges from 0 to 4.

### 2.3. Online Information-Seeking Behavior

In relation to digital health information sources, we asked about the usage of 6 digital health information sources, including official platforms: official media (WHO; ministry of health), health professional websites, search engines (like Google, Yahoo), Wikipedia, and other online encyclopedias, general websites, and blogs. Each of the online sources could be answered in relation to their frequency, including the following response categories: never, rarely, sometimes, and often. For all analyses, response options never and rarely were merged in one option, and rarely and sometimes in another option.

### 2.4. Ethical and Legal Requirements

The study protocol was assessed and approved by the Ethical Committee of Universidade Católica Portuguesa (number 207; CES). In this investigation, we followed the principles of the Helsinki Declaration and the Oviedo Convention about the protection of human rights in biomedical investigation. All students formally consented to participate in the study.

### 2.5. Statistical Analysis

Descriptive analysis was carried out on the variables used to characterize the sample. Continuous variables were described by mean and standard deviation, and categorical variables by absolute frequency and relative frequency (%).

We performed bivariate analyses by using the chi-squared test to determine whether there are any differences between participant sociodemographic characteristics, and knowledge and the two levels of digital health literacy (limited vs. sufficient).

A binary logistic model was made to identify the factors associated with a sufficient level of e-health literacy. Multivariate analysis considered adjustments for age, sex, and academic level. The results are presented in odds ratio (OR) and respective 95% confidence interval.

The statistical analysis of the data was performed in Statistical Package for the Social Sciences (SPSS, version 27.0, IBM, Chicago, IL, USA). The level of significance was set to 0.05.

## 3. Results

In this study, 534 college students participated; the majority were female 410 (76.8%). The mean age was 24.3 (SD = 7.8) years, the mean e-health score was 28.8 (5.6). Most of the students assessed reported that the current income allows them to live comfortably (83.7%). Most of the students were attending an undergraduate course, with 24.9% on a master’s course and 8.1% on a doctoral course (Table 1). Regarding the field of study, 53.0% of the students were from non-health area studies and 47.0% (251) were from health area. Regarding the number of hours that students used the Internet, most of them used it more than two hours a day (84.1%); to search general health information in the last 2 month, 42.7% used the Internet for less than 1 h, 25.7% used it between 2 and 3 h, and 14.0% more than 3 h.

### 3.1. E-Health Literacy

The results of the e-health literacy scale for university students are shown in Table 2. In relation to the question “How useful is the Internet in helping you make decisions about your health?”, most students answered it was important, with 56.9%. And around 51.7% of students consider it important to have access to health resources available on the Internet.

Regarding the observed values of the e-health score in our study (Table 3), the statement “I can distinguish between high-quality and low-quality health resources on the Internet” (mean = 3.8) obtained the highest mean and the statement “I feel confident using information from the Internet to make health decisions” the lowest mean score. For the eHEALS scale questions, most of the university students’ answers were in the agree or strongly agree options. The statement “I feel confident using information from the Internet to make health decisions”. Had the lowest percentage of very much agree option.

Students’ field of study, Internet usage, adopting preventive measures, and knowledge score on COVID-19, were significantly associated with e-health literacy (Table 4). The use of online information sources to search for health information was more frequent in students with sufficient than adequate e-health literacy (39.6% vs. 16.4%). In both groups of students, with limited and sufficient e-health literacy, they indicated adherence to respiratory disease prevention behaviors (>50%; *p* = 0.016).

Regarding the knowledge related to COVID-19, the statement “Vaccination for COVID-19 is not a one-time vaccination, it involves boosters” had the highest percentage of correct answers (95.6%) followed by the statement “There is need for time intervals between different courses of the COVID-19 vaccine” (92.9%). The statement “People with fever or a cold cannot be vaccinated with the COVID-19 vaccine” had the lowest percentage of correct answers in our sample (Figure 1). The knowledge score showed a mean value of 4.05 (SD = 1.24) in our sample.

The analyses concerning possible differences between levels of e-health literacy and student characteristics and prevention attitudes are shown in Table 3. The field of study and the search for health information online were associated with the level of health literacy. Health science students had a higher level of e-health literacy than students from other areas; those who searched health information online had a higher level of e-health literacy. Regarding attitudes related with COVID-19 and prevention measures, only the statement “prevent measures for respiratory infections in certain places (e.g., wearing masks, wash hands)” were significantly associated with health literacy (*p* = 0.025). Regardless of the students’ literacy level, the majority agreed that COVID-19 vaccination was very important for their health.

### 3.2. Preferred Sources, Platforms, and Topics for Seeking Information

As shown in Figure 2, preferred platforms among students were (i) official media (WHO; ministry of health), (ii) search engines like Google, Yahoo, and (iii) health professional websites, with 76.6%, 68.0%, and 57.8%, respectively, indicating using these sources “often” or “sometimes”. Additionally, social networks (e.g., Facebook, Instagram) were used often or sometimes. Sites blogs, Wikipedia, and other online encyclopedias were less often used for health information. Official media, health professional websites, and search engines were significantly associated with low or high health literacy (*p* < 0.05).

### 3.3. Multivariate Model

The results of the logistic regression model are presented in Table 5. In the multivariate model, being a health science student (OR = 1.6; CI 95% = 1.1; 2.6), searching for health information on the Internet several times (OR = 1.9; CI 95% = 1.2; 3.4), and searching online health information on professional websites (OR = 2.3; CI 95% = 1.4; 3.9), were significantly associated with the likelihood of having a sufficient level of e-health literacy.

## 4. Discussion

Health literacy and digital health literacy are essential for health promotion and have been a growing concern for many countries over the last two decades. University students, due to their access to information, play an important role in society. To our knowledge, only a few studies have examined e-health literacy among Portuguese university students [8,14,15].

In our study, we found a relatively moderate level of e-health literacy [M = 28.8 in a maximum of 40] in university students. This result was similar to findings in others international studies with university students [12,16,17]. A study with 781 American university students found a mean e-literacy of 30.1 [18]; another study in Iranian medical students found a mean of 28.2 in the e-health score, which is very close to the present study [10]. Several factors may explain the differences found, such as Internet access, cultural values, and socio-economic background; additional studies are needed to understand these differences.

The results of our study demonstrate something common in the health e-literacy literature: confidence in searching for health resources on the Internet does not equate to an ability to analyze the quality of information found or the search for credible information sources [12,19,20]. There was a high prevalence of students answering “yes” to the statement “I know how to find useful health resources on the Internet”, which shows confidence in seeking information, but a low level was observed in the statement “I feel confident using information from the Internet to make health decisions”, which demonstrates the low confidence in making decisions based on information available online. As the Internet continues to be the primary source of online health information, there is a need to develop strategies and pedagogies that increase health literacy skills in students [20]. The e-health literacy and online information sources played a very important role in the prevention and control of COVID-19, due to the social distance and isolation that individuals experienced. These conditions meant that information from health professionals did not happen face-to-face. We observed a significant relationship between e-health literacy and the knowledge and attitudes regarding post-COVID-19 information in our study. Previous studies have shown that knowledge and e-health literacy are significantly associated [11,21]. These results are in line with the findings of our study.

In the present study, students with a higher level of e-health literacy used more sources of health information such as official websites and health professional websites than students with lower literacy. Social networks [e.g., Google] were more often used by students with a lower level of literacy to search for health-related information. Similar results were reported by other countries, in studies in Portugal [14], Germany [22] and Slovenia [23], Denmark [24]. Rosário et al. (2020) [14] found the same trend in Portuguese university students; students with better digital health literacy looked for more information in more credible sources such as official sources and health professional websites.

Regarding the comparison between health science students and non-health science students, it was shown that health science students performed better than the others on e-health literacy. This study’s results are consistent with other studies. For example, some studies reveal that students of nursing area, medical students, had higher literacy skills [11,16,25,26,27,28]. These results are not surprising since students that are in the field of health have more academic experience with health-related content.

There are several limitations to consider when interpreting the results of our study. First, the results were from cross-sectional survey, so we cannot establish causality among the variables; and we used a convenience sample of university students, so it is not representative of Portuguese university students. Second, as university students are the most educated part of society, with the most access to information of various levels, the results are not transposable to society at large. We therefore suggest conducting additional studies but with larger samples, with students from different universities, and with young people of the same age but who are not attending university to assess their digital health literacy.

Regarding the strengths of this study, we would highlight the fact that it is one of the first studies on digital health among Portuguese university students; it took place in a university environment after the COVID-19 period, which greatly influenced the search for health information online. It allows us to find out which sources of information are most used by students and whether they have difficulty choosing between the available sources. To our knowledge, this is the first study on digital health and the search for online health information sources since the COVID-19 period.

Further studies with other dimensions of health literacy and different instruments are needed to clarify some questions on whether access to health information in practice is reflected in better health and health promotion, for example. Intervening with young adults who are attending university as well as those of the same age but who are not in an academic context, promoting their digital health literacy skills, will enable young people to have the tools to search, evaluate, select, and use digital health information. This effort will increase their confidence in this process. It is also suggested to invest in health education in universities in different fields of study [e.g., social sciences, economic, law] and make efforts to improve health literacy at the university level.

## 5. Conclusions

The present study concludes that e-health literacy is associated with online information seeking and not all university students have a good level of health literacy despite having access to online information resources. Therefore, it is necessary from a public health point of view to develop programs that increase health literacy, and health promotion in students.

## Figures and Tables

**Figure 1 ijerph-21-00271-f001:**
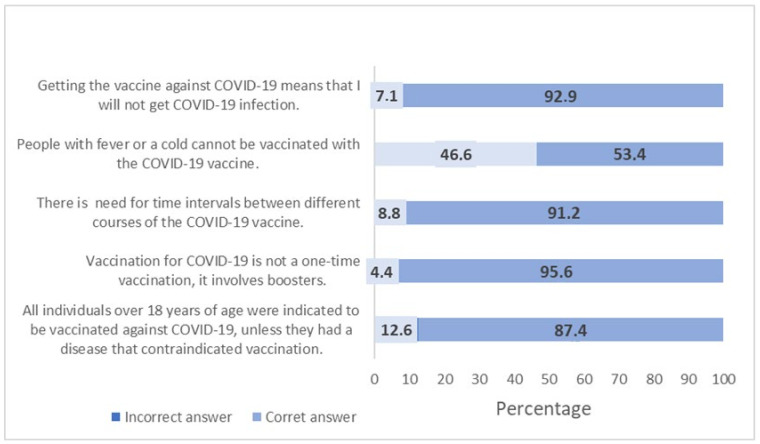
Frequency of correct and incorrect answers about knowledge about COVID-19.

**Figure 2 ijerph-21-00271-f002:**
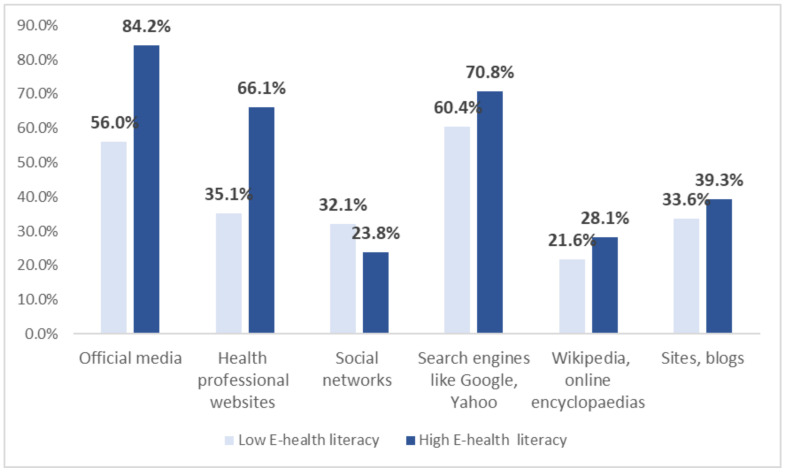
Health information sources according to e-health literacy. Official media, health professional websites, and search engines (google, yahoo) were significant (*p* < 0.05).

**Table 1 ijerph-21-00271-t001:** Descriptive characteristics of university students (N = 534).

	N	%
Age (mean, SD)	24.3 (7.8)	
e-health score (mean, SD)	28.8 (5.6)	
**e-health**		
Inadequate	45	8.4
Problematic	101	18.9
Sufficient	388	72.7
**Sex**		
Men	122	22.8
Woman	410	76.8
I prefer not to say	2	0.4
**Year attended**		
1 year	101	18.9
2 year	125	23.4
3 year	88	16.5
4 year	44	8.2
Master	133	24.9
PhD	43	8.1
**Number of hours you use the Internet**		
Less than 1 h/day	5	0.9
1–2 h/day	80	15.0
>2 h/day	449	84.1
**Search for general health information in the last two months, in total**		
Less than 1 h	228	42.7
1–2 h	137	25.7
2–3 h	60	11.2
More than 3 h	75	14.0
**Health vs. non-health**		
Non-health area	283	53.0
Health area	251	47.0
**Income**		
The current income allows us to live comfortably.	447	83.7
It is difficult to live on current income.	73	13.7
Cannot judge/would rather not answer.	14	2.6

**Table 2 ijerph-21-00271-t002:** The e-health scores for the e-literacy scale for the university students (n = 534).

	N	%
**1. How useful is the Internet in helping you make decisions about your health?**		
1—Totally useless	12	2.2%
2—Useless	48	9.0%
3—Not sure	94	17.6%
4—Useful	304	56.9%
5—Very useful	76	14.2%
**2. How important do you consider to have access to health resources available on the Internet?**		
1. Nothing important	3	0.6%
2. Not important	13	2.4%
3. Not sure	36	6.7%
4. Important	276	51.7%
5. Very important	206	38.6%

**Table 3 ijerph-21-00271-t003:** Percentage of e-health scores for the e-literacy scale for the university students (n = 534).

E-Heals Item	Strongly Disagree	Disagree	I am Not Sure	Agree	Strongly Agree	M
3. I know what health resources are available on the Internet.	1.5%	9.4%	30.7%	49.1%	9.4%	3.55
4. I know where to find useful health pages on the Internet.	1.7%	9.6%	17.8%	56.6%	14.4%	3.72
5. I know how to find useful health resources on the Internet.	1.9%	8.8%	18.4%	56.4%	14.6%	3.73
6. I know how to use the Internet to answer my health questions.	2.2%	8.8%	22.7%	52.4%	13.9%	3.67
7. I know how to use the health information I find on the Internet to help me.	1.7%	6.9%	21.7%	55.8%	13.9%	3.73
8. I can evaluate the health resources I find on the Internet.	2.1%	11.0%	23.2%	51.9%	11.8%	3.60
9. I can distinguish between high-quality and low-quality health resources on the Internet.	2.1%	7.9%	21.2%	49.1%	19.9%	3.77
10. I feel confident using information from the Internet to make health decisions.	8.2%	23.8%	30.1%	29.8%	8.1%	3.06

M = mean.

**Table 4 ijerph-21-00271-t004:** Comparison of observed e-health literacy scores by student characteristics.

	Limited E-Health		Sufficient E-Health		
	n	%	n	%	*p*
**Sex**					
Male	33	27.0	89	73.0	0.953
Female	112	27.3	298	72.7	
**Year graduate**					
Undergraduate	107	29.9	251	70.1	0.103
Masters	32	24.1	101	75.9	
PhD	7	16.3	36	83.7	
**Field of study**					
Health	50	19.9	201	80.1	<0.001
Non-health	96	33.9	187	66.1	
**E-sources for general health information**					
Less than 1 h/day	83	42.3	113	57.7	<0.001
1–2 h/day	29	21.2	108	78.8	
More than 3 h/day	22	13.2	145	86.8	
**Vaccination Likeliness**					
Disagree	49	29.5	117	70.5	0.333
Agree	85	25.4	249	74.6	
**There is need for vaccines against COVID-19 in the future**					
Disagree	40	25.3	118	74.7	0.611
Agree	94	27.5	248	72.5	
**Social distancing (hand washing)**					
Disagree	65	28.6	162	71.4	0.418
Agree	69	25.3	204	74.7	
**Preventive measures (respiratory diseases)**					
Disagree	26	40.0	39	60.0	0.016
Agree	108	24.8	327	75.2	

**Table 5 ijerph-21-00271-t005:** Multivariate logistic regression model of factors associated with e-health literacy (e-health); the dependent variable is sufficient e-health literacy.

	OR	95%CI	*p*
Age	1.03	1.01; 1.07	0.040
**Sex**			
Masculine	1		
Feminine	0.86	0.52; 1.44	0.570
**Health vs. non-health area**			
Non-health area	1		
Health area	1.60	1.04; 2.64	0.032
**Internet use general (day)**			
Less than two hours/day	1		
More than two hours/day	1.43	0.788; 2.62	0.237
**Internet use for health information**			
Less than 1 h	1		
1–2 h	1.67	0.97; 2.88	0.060
more than 3 h	1.90	1.07; 3.42	0.031
**Official media use**			
No	1		
Yes	2.31	1.37; 3.90	0.002
**Health professional websites**			
No	1		
Yes	2.26	1.39; 3.66	<0.001
**Knowledge**			
Low	reference		
High	1.41	0.71; 2.82	0.326

OR—odds ratio. 95%; CI—confidence interval.

## Data Availability

The data presented in this study are available upon reasonable request from the corresponding author.

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
