# Peer review of "Evaluating E-Health Literacy, Knowledge, Attitude, and Health Online Information in Portuguese University Students: A Cross-Sectional Study"

_ijerph, 2024, doi:10.3390/ijerph21030271_

Round 1
Reviewer 1 Report
Comments and Suggestions for Authors
Dear Authors,
thank you for the opportunity to review your well-written manuscript, and I have only one suggestion. Given that an online questionnaire was used. Can the authors show the response rate or sample size calculation?
Reviewer 2 Report
Comments and Suggestions for Authors
Dear authors. I appreciate the opportunity to have been able to read and review your work. I have to admit that I found it interesting. I would like to send you a series of comments/observations.
- The topic of study is relevant, current and whose study and analysis is important due to the great digitalization in health of our times.
- Applying the eHEALS scale is appropriate and consistent with the objective of the work.
- I would like the work to detail why they have decided to use COVID as a central topic of search and study. What did you base your COVID knowledge questions on? I consider it relevant to write that information in the manuscript.
- How have you decided on the sources to search for health-related information and what were you based on to decide which ones to include?
- What do you think is the reason why 76.8% of the participants were women?
- What quality criteria were used to formulate the following question "I can distinguish between high quality and low-quality health resources on the Internet"?
- They must unify the format of the tables, for example, in table 1 they include % as a symbol but, in table 2 they include, after each figure, the symbol %.
- In Table 1 the average is expressed with one decimal, in Table 2, with two. It is important to unify criteria.
- I agree to the limitations they include. It would be interesting to have read in your manuscript the strengths of your research and the influence that the results may have at a social, educational or public health level.
- I consider that it would be interesting to extrapolate this research to populations that, of similar ages, are not pursuing university studies to analyze whether the higher or lower educational level of the participants can interfere with the results.
Thank you so much
Reviewer 3 Report
Comments and Suggestions for Authors
The article is prepared very carefully.
The authors made a detailed analysis of e-Health Literacy, knowledge, attitude, and health
online information in Portuguese university students. Moreover, the study is cross-sectional.
The authors cite many other/similar studies on this topic, which they carefully refer to in the discussion.
Undoubtedly, the limitation of the study is the selection of the study group: all students are from the Catholic University of Portugal (although it has four Campi).
However, several issues were noticed that needed improvement:
1. chapter 3.3. The multivariate model requires correction, numerical data in the text are not always rounded correctly
2. the word "internet" written in lowercase and uppercase letters interchangeably, the wording should be unified
3. figures and tables need to be formatted, full column descriptions are not visible in figures (e.g. 1 and 2)
4. table 4: why did the authors use the form "masculine/feminine" instead of "men/woman"?
5. why is there a "dash" before the descriptions of tables and figures, i.e. "-"?
6. the References requires standardization and adaptation to the editorial requirements
Round 2
Reviewer 2 Report
Comments and Suggestions for Authors
Dear authors. I congratulate you for your work. Thank you for your responses and for taking into account the comments of the different reviewers. From my point of view the manuscript has improved and is of sufficient quality to be published.
Congratulations on your work. I encourage you to continue investigating.
King regards